# Microarray-based analysis reveals a novel role of the miRNA-613/SNAI2/CXCR4 axis in atrial fibrillation

Zhenyu Zhai[1]*, Yiligong Qi[2‡], Longlong Hu[1‡], Zumao Gan[1‡]

1 Department of Cardiovascular Medicine, the Second Affiliated Hospital of Nanchang University, Nanchang, Jiangxi, China, 2 Department of Medical Intensive Care Unit, Jiangxi Maternal and Child Health Hospital, Nanchang, Jiangxi, China

‡ These authors share first authorship on this work.
* 1947344097@qq.com

## Abstract

### Backgrounds

Atrial fibrillation (AF) can lead to substantial morbidity and mortality in clinic. The previous studies demonstrated that miRNAs were closely associated with several cardiovascular diseases, however, the role of miRNAs in the pathogenesis of AF has not been fully elucidated. In order to investigate the important role of miRNA in the mechanisms of AF, we conducted the study through bioinformatics analysis.

### Methods

We downloaded the miRNA expression profile (GSE68475) and mRNA expression profile (GSE31821) from the Gene Expression Omnibus (GEO) database to explore the differentially expressed miRNAs and mRNAs. The criteria for significant differentially expressed miRNA and mRNA using the R limma package were: adjusted P-value < 0.05, log2fold-change ≥ 1. The target mRNAs related to differentially expressed miRNAs of AF were predicted by using Functional enrichment analysis tool. We Screened overlapped mRNAs based on differentially expressed mRNAs and miRNA related mRNAs using Draw Venn Diagram. GO enrichment analysis and KEGG pathway analysis were conducted to explore the role of miRNAs and mRNAs in the pathogenesis of AF.

### Results

A total of 70 differentially expressed miRNAs were screened including 33 up-regulated miRNAs and 34 downregulated miRNAs. All of 94 differentially expressed mRNAs were screened including 56 up-regulated mRNAs and 38 downregulated mRNAs. There were three co-expressed up-regulated differentially expressed genes,

**Data availability statement:** All datasets analyzed in this study are publicly available and can be accessed through the NCBI Gene Expression Omnibus (GEO) database (https://www.ncbi.nlm.nih.gov/geo/). Specifically, the miRNA expression profiling dataset (GSE68475) is accessible at https://www.ncbi.nlm.nih.gov/geo/query/acc.cgi?acc=GSE68475, and the mRNA expression profiling dataset (GSE31821) is available at https://www.ncbi.nlm.nih.gov/geo/query/acc.cgi?acc=GSE31821.

**Funding:** This study was supported by grants from the National Natural Science Foundation of China (No. 82260067), Key Research Project of Jiangxi Provincial Department of Education (GJJ2200105), General Science and Technology Program of Jiangxi Provincial Health Commission (No. 202210627), and Science and Technology Project of Jiangxi Provincial Administration of Traditional Chinese Medicine (No. 2022B903).

**Competing interests:** The authors have declared that no competing interests exist.

including *CXCR4*, *SNAI2*, and *FHL1*. We showed the results of GO functional enrichment analysis and KEGG pathway analysis ranked by enrichment score (-log P value) respectively.

## Conclusion

Compared with patients of normal sinus rhythm, miRNA-613 was significantly down-regulated in patients with AF. We demonstrated that SNAI2 and CXCR4 may target genes of miRNA-613 for the first time. Our findings may provide new ideas for clarifying the molecular mechanism of atrial fibrillation.

## 1. Introduction

Atrial fibrillation (AF) is the most widely diagnosed form of arrhythmia, leading to high clinical morbidity and mortality rates. Both male and female AF patients face a 5-fold increase in their odds of ischemic stroke incidence such that this condition imposes a serious economic burden on healthcare systems and societies throughout the globe [1–4]. AF incidence is particularly common in the context of aging, and the rates of AF diagnosis are also rising rapidly in low- and middle-income nations in which there are limited treatments available such that many cases are unreported, and many patients experience premature death. Given these rising rates of AF, this is a growing public health focus on helping to minimize the associated clinical risk [5,6]. Radiofrequency catheter ablation (RFCA) can be an effective treatment option for patients with AF, particularly among individuals with symptomatic drug-refractory AF [7–9]. For this treatment strategy, the pulmonary veins that are the source of AF-related paroxysmal activity are isolated [9,10]. While RFCA is more effective than drug treatment as a means of preventing AF recurrence, but this intervention can fail such that repeat treatment can be required in 20–50% of patients, and many patients nonetheless face a long-term risk of this condition recurring even after experiencing initial positive clinical outcomes [11,12]. Challenges associated with recurrent AF following ablation procedures remain a significant clinical concern. Despite some advances in the management of AF patients, there is still a pressing need for the identification of new diagnostic and prognostic biomarkers associated with this condition and for the establishment of scientifically-validated treatment options that can reduce the rates of AF and associated sequelae. Efforts to study the mechanistic basis for AF onset and persistence may thus offer insights that can aid in the stratification and treatment of affected patients, potentially by aiding the further optimization of current ablation approaches [13]. Several articles have shown that left atrial remodeling is closely related to the onset of AF, but the mechanistic basis for such remodeling remain to be fully determined [14]. Plasma and tissue microRNAs (miRNAs) have been shown to be differentially expressed in middle-aged and elderly atrial arrhythmia patients, suggesting a potential regulatory role for these miRNAs in atrial arrhythmia development [15–20].

As a kind of non-coding RNA sequences 22–25 nucleotides long, miRNAs serve as important post-translational regulators of gene expression owing to their ability to repress the translation or induce the degradation of complementary target RNAs [21–23]. Roughly 2,200 evolutionarily conserved miRNAs have been characterized to date in humans, regulating a diverse array of targets such that they serve as important regulators of a wide range of both pathological and physiological processes [24–27]. Given their ability to regulate vital cardiovascular processes, the dysregulation of miRNAs in vascular tissue or systemic circulation may be linked to the pathogenesis of cardiovascular conditions such as hypertension, cardiomyopathy, atherosclerosis, heart failure, ischemic heart disease, and stroke. However, few reports to date have explored the association between miRNAs and AF in detail, although some reports have suggested that certain miRNAs may offer value as diagnostic or prognostic biomarkers associated with this condition [28,29]. Recent advances in RNA-sequencing, microarray, and bioinformatics technologies have enabled the comprehensive evaluation of miRNAs and the characterization of their ability to silence mRNA expression through destabilization, cleavage, poly(A) tail shortening, and the impairment of ribosome-mediated translation efficiency [30–33]. While there have been several studies of a range of miRNAs over recent years, the roles of many of these miRNAs in humans have yet to be established [34–36]. Although several recent clinical analyses have also suggested that certain circulating miRNAs specific to the cardiovascular system may offer diagnostic or therapeutic value, additional work is required to translate these findings to the clinic. Accordingly, further work assessing the role that miRNAs play in AF development is warranted to guide the detection of new diagnostic biomarkers and therapeutic targets that can help alleviate the burden of AF.

To explore the role of miRNAs as regulators of AF development, bioinformatics analyses were performed using two publicly available AF-related mRNA and miRNA expression datasets from the Gene Expression Omnibus (GEO) database (GSE68475 and GSE31821). Functional enrichment analyses were then used to explore the potential functions of the target genes of miRNAs differentially expressed in AF samples. Co-expressed genes significantly differentially expressed between these two groups were then analyzed, ultimately revealing that miR-163 was downregulated in AF patient samples whereas putative miR-163 target genes including SNAI2 and CXCR4 were upregulated in these patients as compared to control individuals. These results may offer new insight into the pathogenesis of AF, providing a foundation for future efforts to detect biomarkers associated with the diagnosis of AF and to design novel treatments for this debilitating disease.

## 2. Materials and methods

### 2.1. AF-related miRNA expression profiling

An AF-related miRNA microarray dataset (GSE68475) was downloaded from the NCBI GEO database (NCBI, http://www.ncbi.nlm.nih.gov/geo/). This dataset included miRNA expression levels in right atrial appendage samples collected from individuals with AF or normal sinus rhythm (NSR) undergoing open-heart surgery at Oita University Hospital. In total, this dataset included 10 AF patient samples and 11 NSR patient samples. Raw microarray data were preprocessed and analyzed using GeneSpring GX software. Briefly, raw data values below 1.0 were set to 1.0 to minimize potential interference from very low-expression miRNAs. Next, the data for each miRNA were normalized to the median expression level of that miRNA across the 11 NSR samples, eliminating inter-array systematic errors and technical bias. Following normalization, we applied additional filtering to retain only miRNAs with raw signal intensities of ≥20 in at least one sample, ensuring the reliability of subsequent analyses. Principal component analysis (PCA) was performed as a quality control measure, and results demonstrated clear separation between AF and NSR groups, supporting the suitability of the dataset for differential expression analysis.

### 2.2. Differentially expressed miRNA identification

Differentially expressed miRNAs between AF and NSR patient samples were identified using the R limma package. Briefly, moderated t-tests were performed to compare miRNA expression between the two groups, and P values obtained

from these tests were subsequently adjusted for multiple comparisons using the Benjamini-Hochberg False Discovery Rate (FDR) method. MiRNAs with adjusted P-value < 0.05 were considered statistically significant and selected for further analyses.

## 2.3. Functional enrichment analyses

Gene ontology (GO) and enrichment analysis and Kyoto Encyclopedia of Genes and Genomes (KEGG) enrichment analyses of the miRNAs of interest were performed with the FunRich tool (v 3.1.3). Analyzed GO terms included those in the biological process (BP), cellular component (CC), and molecular function (MF) categories. KEGG pathway analyses provide insight into the pathways associated with particular targets, diseases, or other biological processes of interest [37].

## 2.4. Identification of overlapping differentially expressed miRNAs and mRNAs associated with AF

The GSE31821 dataset was downloaded from the GEO database and included auricular tissue biopsy samples from 2 control individuals and 4 AF patients from which RNA were extracted for Affymetrix microarray analysis. Raw microarray data were imported into R and preprocessed using the Robust Multi-array Average (RMA) method, including background correction to reduce background noise, quantile normalization to eliminate technical variability among arrays, and probe summarization to obtain gene-level expression values. Principal component analysis was performed as a quality control measure, revealing clear separation between AF and control samples, supporting data reliability for subsequent analyses. The target mRNAs related to differentially expressed miRNAs of AF were predicted by using Functional enrichment analysis tool (FunRich, version 3.1.3). We Screened overlapped mRNAs based on differentially expressed mRNAs and miRNA related mRNAs using Draw Venn Diagram. Those differentially overexpressed mRNAs overlapping with miRNAs identified above were identified using a Venn diagram (http://bioinformatics.psb.ugent.be/webtools/Venn/). The functions of overlapping targets were explored through functional enrichment analyses as above.

## 2.5. Statistical analysis

Bioinformatics analyses were performed with R v3.6.0 and the ActiveState Perl software programming language. Differential miRNA and mRNA expression were analyzed with t-tests in the R limma package, with adjusted P-value < 0.05 as the threshold for significance in these microarray datasets [38].

## 2.6. Experimental model and cell culture

Human ventricular cardiomyocytes (AC16; Sigma-Aldrich, Cat# SCC-AC16) were cultured in DMEM/F-12 (Gibco, Cat# 11320−033) supplemented with 10% fetal bovine serum (FBS; Gibco, Cat# 10099−141) and 1% penicillin-streptomycin (Gibco, Cat# 15140−122) at 37 °C in a humidified atmosphere containing 5% $CO_2$. Cells were maintained mycoplasma-free, authenticated by short-tandem-repeat profiling, and used between passages 4–10 to ensure phenotypic stability. For all experiments, cells were seeded at $2 \times 10^5$ cells well$^{-1}$ in collagen-I-coated 6-well plates (Corning) and allowed to reach ~70% confluence before transfection.

## 2.7. miRNA mimic/inhibitor transfection

Gain- and loss-of-function studies were performed with synthetic hsa-miR-613 or hsa-miR-339-5p mimics, inhibitors, and matched negative controls (RiboBio, Guangzhou). Oligonucleotides were transfected at a final concentration of 50 nM using Lipofectamine 3000 (Thermo Fisher Scientific, Cat# L3000-015) according to the manufacturer's instructions. Briefly, Lipofectamine and oligonucleotides were separately diluted in Opti-MEM (Gibco), combined (1:1 vol/vol), incubated for 15 min at room temperature, and added dropwise to the culture medium. Six hours later, the medium was replaced with fresh growth medium. Cells were harvested 48 h post-transfection for RNA extraction.

## 2.8. RNA isolation and quantitative reverse transcription PCR

Total RNA was extracted with TRIzol™ Reagent (Thermo Fisher Scientific, Cat# 15596−018) and quantified spectrophoto-metrically (NanoDrop 2000; Thermo Fisher). RNA integrity was verified by agarose-gel electrophoresis. For mRNA analysis, 500 ng RNA were reverse-transcribed with the PrimeScript™ RT Reagent Kit (Takara, Cat# RR037A) using random hexamers; for miRNA analysis, 200 ng RNA were reverse-transcribed with the Mir-X™ miRNA First Strand Synthesis Kit (Takara, Cat# 638313) employing stem-loop primers. Quantitative PCR was performed on a CFX96 Touch™ Real-Time PCR Detection System (Bio-Rad) with TB Green® Premix Ex Taq™ II (Takara, Cat# RR820A). The 20 μL reaction comprised 10 μL 2×TB Green mix, 0.4 μL of each 10 μM primer, 1 μL cDNA, and 8.2 μL nuclease-free water. Cycling parameters: 95 °C 30 s; 40 cycles of 95 °C 5 s and 60 °C 30 s; followed by melt-curve analysis. Primer sequences for GAPDH, FHL1, CXCR4, SNAI2, and U6 (miRNA reference) are listed in table shown below:

**Primer sequences used for qPCR**

| Target | Forward (5'→3') | Reverse (5'→3') |
|---|---|---|
| GAPDH | GGAAGCTTGTCATCAATGGAAATC | TGATGACCCTTTTGGCTCCC |
| FHL1 | TGCTGCCTGAAATGCTTTGAC | GCCAGAAGCGGTTCTTATAGTG |
| CXCR4 | GGGCAATGGATTGGTCATCCT | TGCAGCCTGTACTTGTCCG |
| SNAI2 | GAACTGGACACACATACAGTGATT | GGCTGTATGCTCCTGAGCTG |

## 2.9. Data analysis

Ct values were averaged from three technical replicates per sample. Relative expression was calculated using the $2^{-\Delta\Delta Ct}$ method with GAPDH (mRNA) or U6 (miRNA) as internal controls and the corresponding negative-control group (Mimic-NC or Inhibitor-NC) as calibrator. Each experiment was repeated independently three times (n=3). Data are presented as mean±SD. Statistical significance between two groups was assessed by unpaired two-tailed Student's t-test in GraphPad Prism 9; $p < 0.05$ was considered significant.

## 3. Results

### 3.1. Differentially expressed AF-related miRNA identification

An initial analysis of 1,023 miRNAs in a previously published microarray dataset including samples from 10 persistent AF patients and 11 NSR controls was performed, with expression levels in different samples being normalized prior to analysis such that the median values were consistent across all patients (Fig 1A&B). Then, the R limma package was used to identify differentially expressed miRNAs, of which 70 were ultimately detected, including 33 and 34 that were respectively upregulated and downregulated in AF samples as compared to NSR controls. These differential expression profiles were also compared with volcano plots (Fig 1C).

### 3.2. Functional enrichment analyses of miRNAs of interest

To evaluate the possible biological effects of the identified differentially expressed miRNAs, miRNAs of interest were subjected to GO, KEGG, and transcription factor enrichment analyses with the FunRich tool (v 3.1.3). For the GO analysis, enriched terms were ranked based on enrichment score values (-logP-value) (Fig 2). The top 3 enriched biological process (BP) terms for these miRNAs were signal transduction (24.8%, P>0.05), regulation of nucleobase, nucleoside, nucleotide and nucleic acid metabolism (19.4%, P>0.05), and transport (8.6%, P>0.05) (Fig 2A). Dysregulation of signal transduction pathways is widely recognized as a crucial factor in AF pathogenesis due to their roles in modulating cardiac electrophysiological properties, ion channel activities, and cellular responses. Aberrant nucleotide metabolism and transport processes may influence cellular energy metabolism, transcriptional regulation, and RNA processing, contributing to atrial remodeling and structural alterations frequently observed in AF patients. Regarding cellular components (CC), the top three enriched

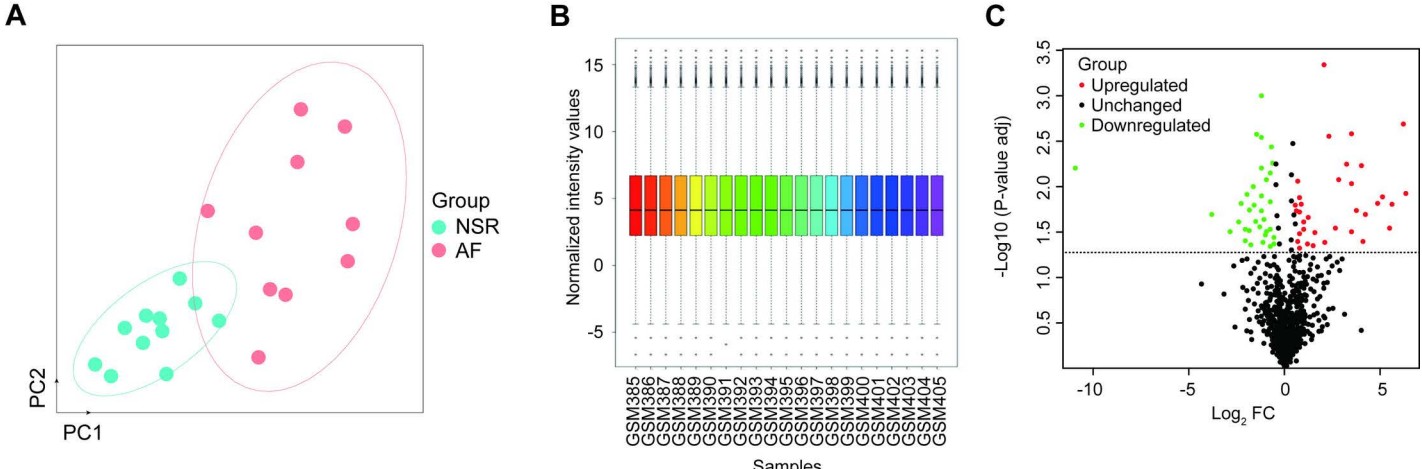

**Fig 1. Identification of differentially expressed atrial fibrillation-related miRNAs. (A)** PCA plot of the miRNA expression profiles from dataset GSE68475, showing clear separation between normal sinus rhythm (NSR, blue dots) and atrial fibrillation (AF, red dots) samples. Ellipses represent 90% confidence intervals for each group. **(B)** The box plot showed the normalized expression values of miRNAs between atrial fibrillation group (GSM385-GSM395) and normal sinus rhythm group (GSM396-GSM405). **(C)** Volcano plot displaying the differentially expressed miRNAs identified between AF and NSR samples. Red dots indicate significantly upregulated miRNAs, green dots represent significantly downregulated miRNAs, and black dots represent miRNAs without significant changes.

terms were nucleus (51.8%, P<0.001), cytoplasm (49.3%, P<0.05), and lysosome (17.4%, P<0.05) (Fig 2B). Enrichment of miRNAs targeting genes predominantly located in the nucleus and cytoplasm aligns with their known roles in gene expression regulation, transcriptional control, and signal transduction, which are all relevant to AF pathophysiology. The significant enrichment of lysosomal components suggests potential involvement of autophagy-lysosome pathways in atrial remodeling and fibrosis, mechanisms known to underlie AF development and progression. Additionally, the top three molecular function (MF) terms were transcription factor activity (7.8%, P>0.05), transporter activity (5.6%, P>0.05), and RNA binding (4.2%, P>0.05) (Fig 2C). Altered transcription factor activity and RNA binding processes can directly modulate gene expression programs implicated in AF, such as fibrosis-associated signaling and inflammatory responses. Similarly, transporter activity changes might influence ion channel functions and intracellular signaling, thus directly contributing to AF-associated electrophysiological disturbances. KEGG pathway analysis results were similarly ranked based on enrichment score values (Fig 2D), revealing that these differentially expressed AF-related miRNAs were most closely associated with proteoglycan syndecan-mediated signaling events (38.6%, P<0.001), syndecan-1-mediated signaling events (37.2%, P<0.001), the TRAIL signaling pathway (37.2%, P<0.001), and the sphingosine 1-phosphate (S1P) pathway (36.7%, P<0.001). Syndecan signaling events are known to regulate cellular adhesion, migration, inflammation, and fibrosis—all central elements involved in atrial remodeling processes observed in AF patients. The TRAIL signaling pathway, typically involved in apoptosis and inflammation, may reflect increased apoptotic and inflammatory activities in atrial tissues affected by AF. Additionally, the S1P signaling pathway is crucial in regulating cardiac electrophysiological stability, cellular proliferation, and migration, suggesting its potential role in AF pathology via promoting structural and electrical atrial remodeling. A transcription factor enrichment analysis was also performed for these miRNAs (Fig 3).

### 3.3. Validation of differentially expressed miRNA target gene expression patterns in a separate AF dataset

A separate GEO dataset (GSE31821) was next analyzed to identify mRNAs that were differentially expressed in AF samples relative to NSR controls, ultimately leading to the detection of 94 differentially expressed genes (adjusted P-value<0.05, Log$_2$FC≥1), of which 56 and 38 were respectively upregulated and downregulated in the AF group (Fig 4).

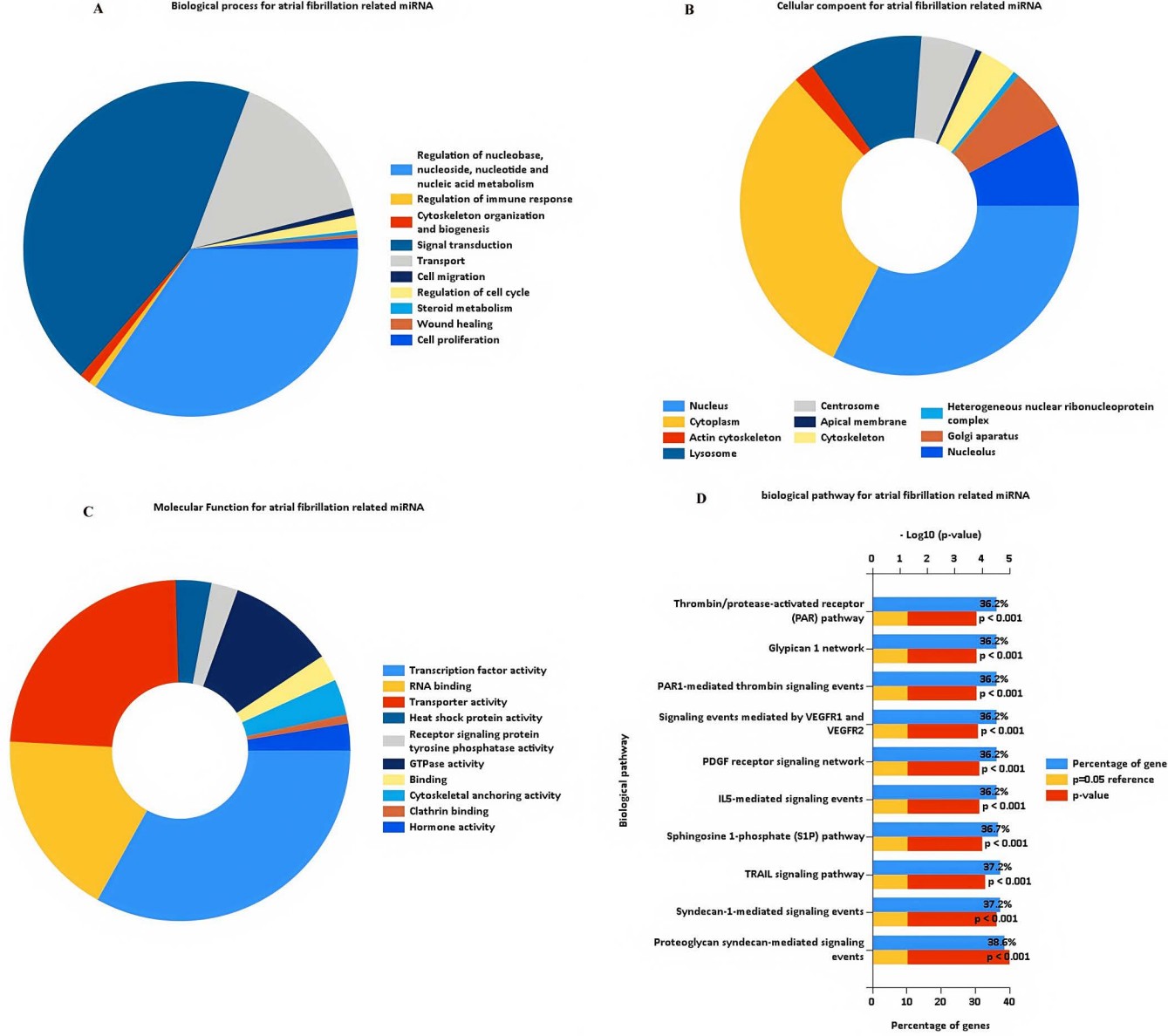

**Fig 2. GO and KEGG enrichment analyses for differently expressed miRNAs in atrial fibrillation group with the FunRich tool (v 3.1.3).** The plots showed GO enrichment analysis for differentially expressed miRNAs related to atrial fibrillation, including biological processes (BP), cellular component (CC), and molecular functions (MF) terms **(A-C)**. The plots showed KEGG enrichment analysis for differentially expressed miRNAs related to atrial fibrillation **(D)**. GO: Gene ontology; KEGG: Kyoto Encyclopedia of Genes and Genomes.

Most miRNAs are capable of binding to complementary sequences for multiple mRNA targets, and the FunRich tool was thus used to predict target mRNAs associated with differentially expressed AF-related miRNAs, leading to the identification of 512 such candidate target genes. A Venn diagram was then used to assess the overlap between these 512 candidate genes and the 94 differentially expressed genes identified in AF samples above (Fig 5). This ultimately identified three overlapping upregulated differentially expressed genes present in both of these datasets (Table 1), including SNAI2, CXCR4, and FHL1. Functional enrichment analyses showed that these genes were enriched in the sensory perception

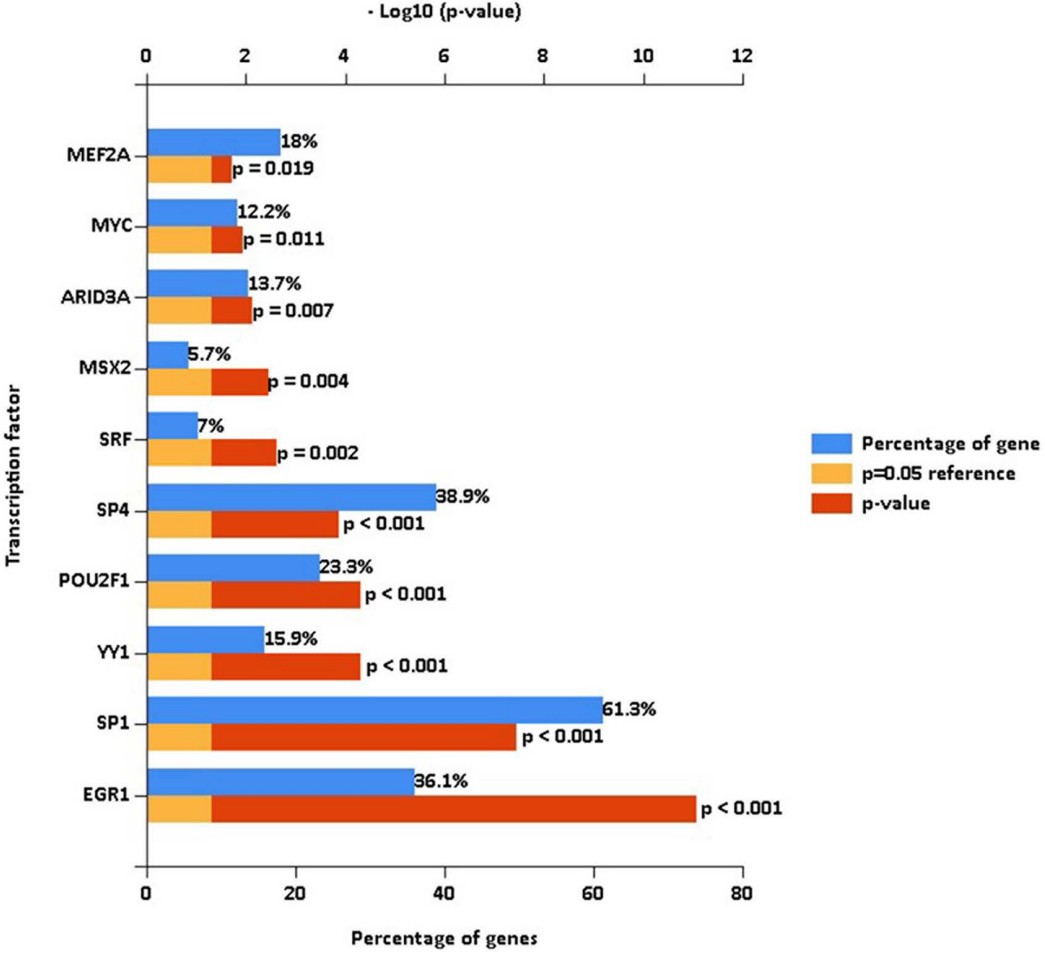

**Fig 3. Transcription factor enrichment analyses for differently expressed miRNAs in atrial fibrillation group with the FunRich tool.**

of mechanical stimulus, regulation of metal ion transport, regulation of cell growth, and cell growth BP terms, while no enriched CC terms were detected suggesting these genes to be widely distributed throughout cells; Low levels of MF term enrichment with similar gene ratio values were also observed for these target genes (Fig 6). A miRNA-mRNA network was further constructed to highlight the association between these mRNAs and their associated miRNA regulators (Fig 7).

### 3.4. Experimental validation of miRNA-mRNA interactions via quantitative PCR

To experimentally validate the predicted miRNA-mRNA regulatory relationships derived from bioinformatics analyses, quantitative polymerase chain reaction (qPCR) assays were performed. Specifically, we investigated the direct effects of miR-339-5p and miR-613 on the expression levels of their predicted target genes (FHL1, SNAI2, and CXCR4) in vitro (Fig 8).

The qPCR results demonstrated that the expression of FHL1 significantly increased when cells were treated with the hsa-miR-339-5p inhibitor compared to the inhibitor negative control (Inhibitor-NC) group (P < 0.0001), indicating a strong negative regulatory relationship. Conversely, treatment with the miR-339-5p mimic markedly decreased the FHL1

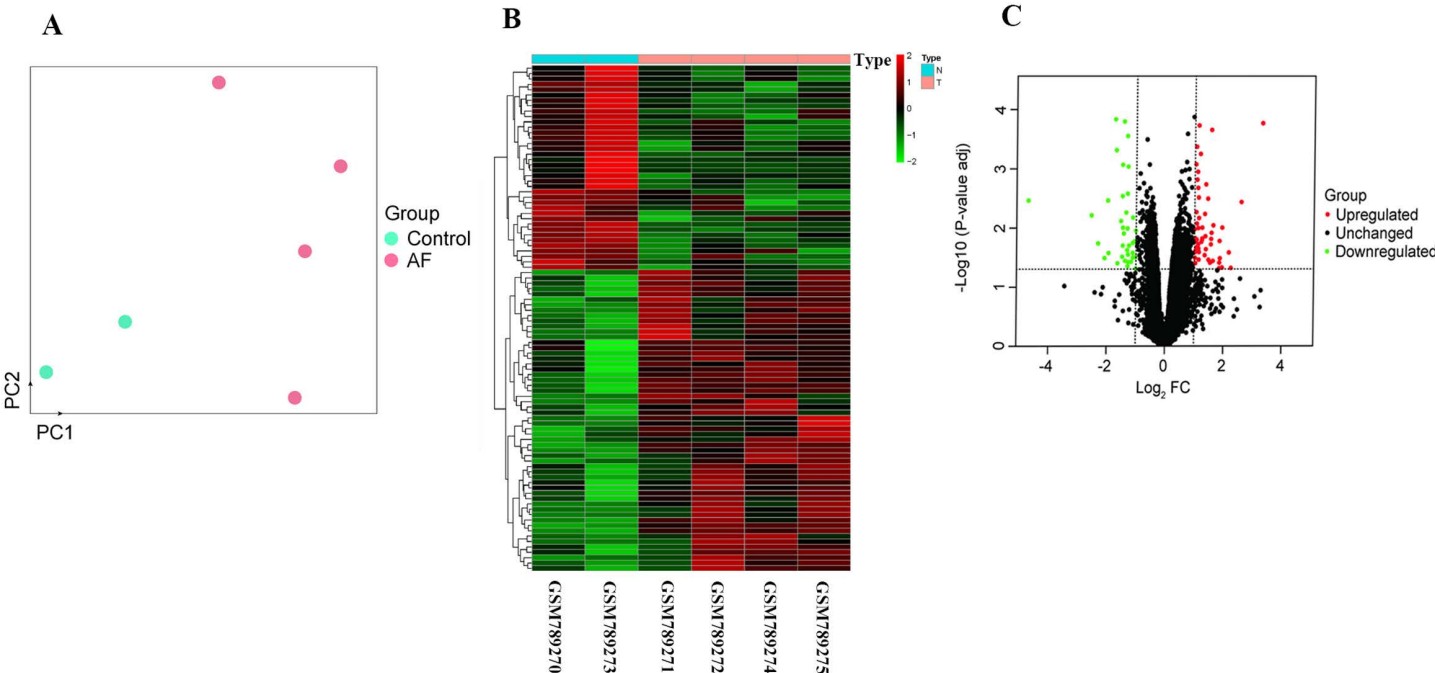

**Fig 4. Identification of differentially expressed atrial fibrillation-related mRNAs. (A)** PCA plot of the mRNA expression profiles from dataset GSE31821, demonstrating distinct grouping of AF patient samples (red dots) and control samples (blue dots), confirming suitable data quality for further analyses. **(B)** The heat map showed the differentially expressed mRNAs in atrial fibrillation group, compared to the normal sinus rhythm group. T indicated the atrial fibrillation group; N indicated the normal sinus rhythm group. Green: represented low relative expression; Red: represented high relative expression. **(C)** Volcano plot illustrating the differential expression of mRNAs between AF and control samples. Similarly, red and green dots indicate significantly upregulated and downregulated mRNAs, respectively.

expression levels compared with the mimic negative control (Mimic-NC) group (P<0.01). These findings clearly support the prediction that miR-339-5p negatively regulates FHL1, reinforcing the complexity of miRNA-mRNA interactions in atrial fibrillation.

Similarly, our analysis confirmed that miR-613 significantly modulated the expression of its putative target genes, SNAI2 and CXCR4. Inhibition of miR-613 led to a substantial upregulation of SNAI2 (P<0.0001) and CXCR4 (P<0.01), compared with the Inhibitor-NC group. In contrast, overexpression of miR-613 using a miR-613 mimic significantly decreased the expression of SNAI2 (P<0.01) and CXCR4 (P<0.001) compared with the Mimic-NC group. These experimental validations confirm the bioinformatics predictions and substantiate our hypothesis that the downregulation of miR-613 observed in atrial fibrillation patients could contribute to disease progression by relieving repression of SNAI2 and CXCR4 expression.

Collectively, these qPCR validation results provide robust experimental evidence confirming the predicted regulatory interactions and strengthen the mechanistic insights into miRNA involvement in atrial fibrillation pathogenesis.

## 4. Discussion

While many prior studies have explored the etiological basis for AF onset and progression, the precise molecular underpinning for this condition remains incompletely understood [39–43]. Non-coding RNAs have been shown to serve as important regulators of AF pathogenesis, with circular RNAs (circRNAs), for example, controlling relevant processes such as cell function, development, and pathological responses relevant to heart disease [44–46]. In contrast to the relatively

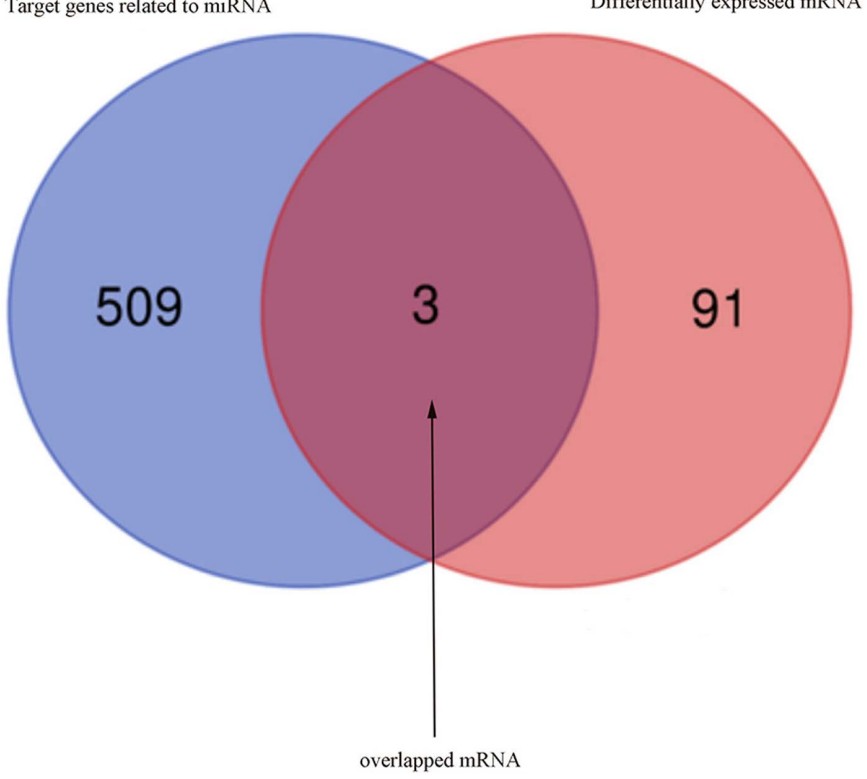

Target genes related to miRNA

Differentially expressed mRNA

overlapped mRNA

**Fig 5. Overlapped mRNAs based on target genes of differentially expressed miRNAs and differentially expressed mRNAs in atrial fibrillation group, compared with the normal sinus rhythm group.**

**Table 1. The expression of overlapped mRNAs and their related miRNAs in atrial fibrillation group.**

| miRNA | mRNA | Target | miRNALogFC | mRNALogFC |
|---|---|---|---|---|
| hsa-miR-613 | CXCR4 | target | −1.24 | 1.05 |
| hsa-miR-339-5p | FHL1 | target | 2.86 | 1.38 |
| hsa-miR-613 | SNAI2 | target | −1.24 | 1.11 |

"Target" indicates overlapped mRNAs were the predicted target genes of related miRNAs. FC, fold change; CXCR4, chemokine receptor 4; FHL1, four-and-a-half LIM domain protein 1; SNAI2, snail family transcriptional repressor 2.

complex biogenesis processes and poor interspecies conservation of circRNAs [47], miRNAs are relatively smaller and simpler non-coding RNAs that post-transcriptionally serve to suppress target gene expression [48]. Many of these miR-NAs have been shown to specifically control the proliferation, survival, or differentiation of particular cell types [49–54], and several miRNAs have been reported to be differentially regulated in AF including miR-210, miR-30e, and miR-641 [55–61]. This study leveraged extant microarray datasets to further explore the potential roles of miRNAs as relevant AF-associated biomarkers.

Initial analyses for this study utilized the GSE68475 dataset consisting of 11 NSR and 10 AF patient samples, enabling the identification of 70 miRNAs that were differentially expressed between these two groups of which 33 and 34 were respectively upregulated and downregulated in AF patients. To extend these results, differential mRNA expression between 2 AF patients and 4 NSR patients was assessed using a separate publicly available microarray dataset,

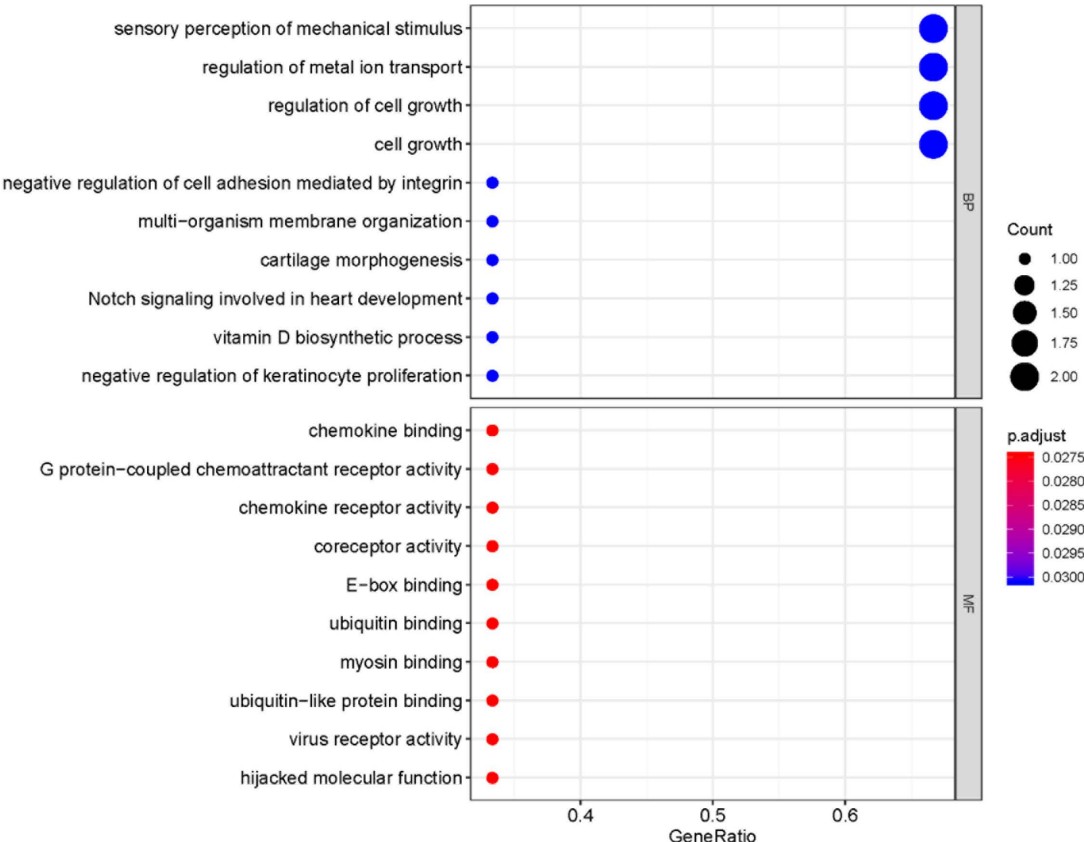

**Fig 6. The bubble chart showed the results of biological process enrichment and molecular function enrichment for overlapped mRNAs in atrial fibrillation group.** The size of the circle indicated the count of enriched mRNAs. Red: represented significant gene enrichment; Blue represented relatively insignificant gene enrichment.

identifying 56 and 38 mRNAs that were significantly upregulated and downregulated in AF samples. By evaluating the overlap between this latter dataset and the putative targets of the 70 miRNAs identified in the first dataset led to the identification of three overlapping genes (SNAI2, CXCR4, and FHL1). While miR-613 was downregulated in AF samples, the consistent upregulation of its target genes SNAI2 and CXCR4 was also observed. In contrast, both miR-339-5p and its target FHL1 were upregulated in AF samples. At the molecular mechanistic level, we hypothesize that the downregulation of miR-613 observed in AF samples may relieve its inhibitory effect on SNAI2 and CXCR4, thus contributing to the enhanced expression of these genes. SNAI2, a zinc finger transcription factor involved in epithelial-mesenchymal transition, may mediate structural remodeling in atrial tissues by promoting fibrosis and extracellular matrix deposition, both critical factors associated with AF development. Likewise, CXCR4, a chemokine receptor, plays essential roles in inflammatory cell recruitment and tissue remodeling—processes known to contribute significantly to atrial fibrosis and the structural remodeling characteristic of AF pathology.

Moreover, the downregulation of miRNA-613 and subsequent upregulation of SNAI2 and CXCR4 suggest that ventricular myocardium might enter a state analogous to myocardial hibernation. Myocardial hibernation refers to a protective remodeling process in which cardiac myocytes downregulate their contractile function and metabolic demand to maintain viability under conditions of chronic ischemia or hemodynamic stress. Typical characteristics include reduced myocardial contractility despite preserved cell viability, glycogen accumulation, decreased capillary density, downregulated metabolic enzyme activity, and activation of ischemia-related signaling pathways [62]. It has been reported

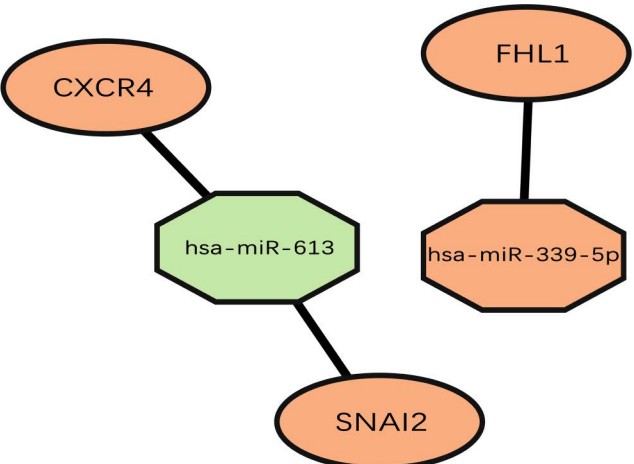

**Fig 7. The miRNA-mRNA network between overlapped mRNAs and their related miRNAs in atrial fibrillation group.** Green: indicated down-regulation in atrial fibrillation group; Orange indicated up-regulation in atrial fibrillation.

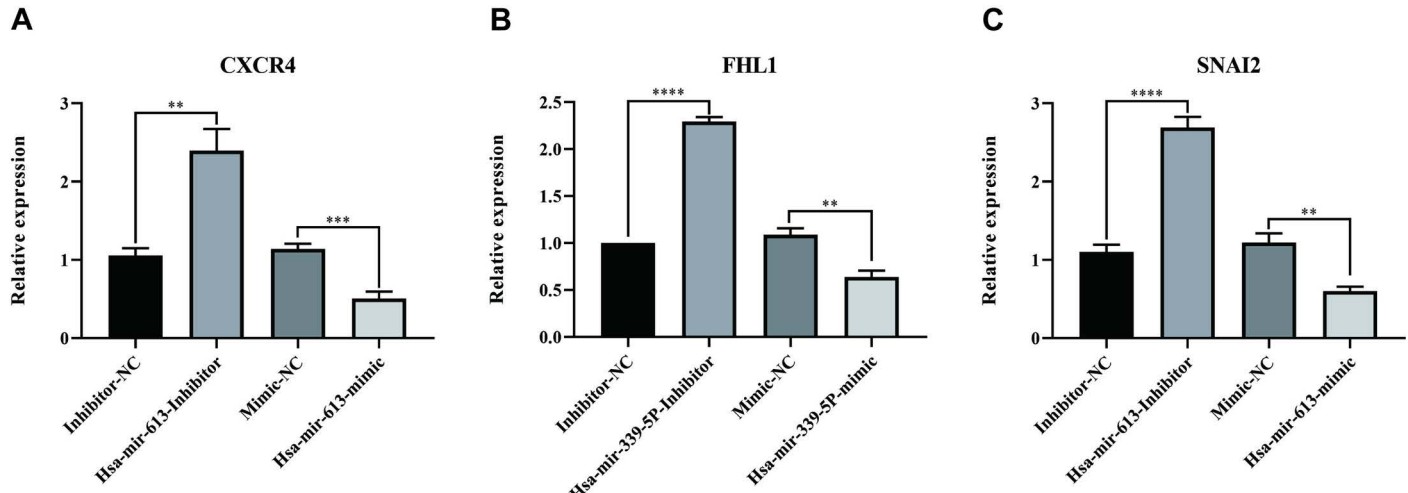

**Fig 8. qPCR validation of miRNA-mRNA regulatory relationships in atrial fibrillation.** Expression level of FHL1 (A), SNAI2 (B) and CXCR4 (C) following transfection with hsa-miR-613 inhibitor or mimic compared to negative controls. Data represent mean±SD of three independent experiments. **P<0.01; ***P<0.001; ****P<0.0001.

that even in the absence of overt coronary ischemia, persistent rapid pacing and ventricular dyssynchrony can induce myocardial changes mimicking hibernation, such as increased glycogen storage and reduced expression of glycolytic enzymes, representing an adaptive response to sustained hemodynamic stress [63]. The transcriptional repressor SNAI2 promotes cellular survival under stress conditions, while elevated CXCR4 expression is considered a hallmark of protective myocardial responses under ischemic conditions. Therefore, the observed miRNA-613/SNAI2/CXCR4 axis might function as a molecular modulator facilitating a hibernation-like phenotype in ventricular myocardium under chronic AF-induced hemodynamic stress, ultimately protecting myocardial tissue from energetic depletion and stress-induced damage.

FHL1 is associated with cytoskeletal dynamics and structural integrity in cardiomyocytes; thus, its upregulation might reflect compensatory or maladaptive remodeling responses occurring during AF progression. These proposed regulatory relationships underline the potential significance of these miRNA-mRNA interactions in the pathogenesis and maintenance of AF. To experimentally validate these bioinformatics predictions, we performed qPCR experiments to assess the direct regulatory relationships between these miRNAs and their predicted target genes. The results clearly demonstrated that inhibition of miR-613 significantly increased the expression of SNAI2 and CXCR4, while the miR-613 mimic markedly decreased their expression. Similarly, miR-339-5p inhibitor significantly elevated FHL1 expression, whereas the miR-339-5p mimic reduced its expression. These qPCR validation experiments strongly support our hypothesis regarding the involvement of miR-613 in AF pathogenesis through negative regulation of SNAI2 and CXCR4, as well as indicating a direct negative regulatory relationship between miR-339-5p and FHL1.Enrichment analyses of the differentially expressed miRNAs revealed that they were enriched in the signal transduction, regulation of nucleobase, nucleoside, nucleotide, and nucleic acid metabolism, and transport BP terms, the nucleus, cytoplasm, and lysosome CC terms, and the transcription factor activity, transporter activity, and RNA binding MF terms. They were further enriched in the proteoglycan syndecan-mediated signaling events, syndecan-1-mediated signaling events, TRAIL signaling pathway, and S1P pathway KEGG terms. Of particular interest, the TRAIL signaling pathway and S1P signaling pathway have previously been implicated in cardiovascular remodeling processes, inflammation, and apoptotic regulation, suggesting their potential involvement in atrial structural remodeling and electrical instability observed in AF. The syndecan-mediated signaling pathways are also noteworthy, given their known roles in extracellular matrix remodeling and cellular adhesion, both crucial factors influencing cardiac fibrosis and the atrial remodeling processes underlying AF pathogenesis. The most enriched BP terms for the three overlapping target mRNAs were sensory perception of mechanical stimulus, regulation of metal ion transport, regulation of cell growth, and cell growth. Notably, regulation of metal ion transport and cell growth are processes closely related to cellular electrophysiology and myocardial structural remodeling, both of which are critical events in the progression of AF.

Several articles published recently have documented the ability of miR-613 to influence oncogenesis and other diseases. MALAT1 is a lncRNA that has been reported to serve as an inhibitor of miR-613 in hepatocellular carcinoma cells, wherein this miRNA can drive tumor metastasis via the peripheral vascular system. Li et al. further found a role for miR-613 as a suppressor of the proliferation of granulosa cells such that it may serve as a regulator of polycystic ovarian syndrome [64]. miR-613 was also identified as a LINC00460 target in LOVO and HT29 cells that in turn targets SphK1 [65], while a separate analysis revealed that overexpressing miR-613 was sufficient to enhance cisplatin chemosensitivity [66].

By promoting fibronectin 1 downregulation and thereby inactivating AKT signaling activity, miR-613 can suppress nasopharyngeal carcinoma cell angiogenic activity [67], and it can further render gastric cancer cells more sensitive to cisplatin treatment by suppressing Sex-determining region Y-box 9 expression [68]. There is further evidence for the ability of miR-613 to suppress Atonal homolog1, promoting colon cancer cell invasivity, migration, and proliferative activity through JNK1 pathway signaling activation and mucin 2 upregulation [69]. In osteosarcoma, miR-613 has also been shown to suppress CXCR4 expression, thereby driving tumor growth and metastasis to the lungs [70].

Overall, these analyses suggested that in AF patients, significant miR-613 downregulation was evident as compared to NSR controls. This miR-613 downregulation may serve to drive the onset or progression of AF via facilitating CXCR4 upregulation. This study is also the first to have identified SNAI2 as a miR-613 target gene candidate that was upregulated in samples from patients with AF. Both miR-339-5p and its putative target gene FHL1 were also herein found to be upregulated in individuals with AF. Overall the limited number of overlapping miRNA target genes and the inconsistent levels of FHL1 in AF samples may be the result of the limited number of samples in the second analyzed microarray (2 NSR and 4 AF samples). Additional research will also focus in greater detail on the mechanisms through which miR-613 and its target genes shape the pathogenesis of this disease, thereby potentially providing a foundation that will enable researchers to better understand the molecular basis for AF and associated treatment options.

There were key strengths to this study. Firstly, the results of our study showed that compared with NSR patients, AF patients exhibited significant miR-613 downregulation for the first time. Secondly, these analyses also revealed that SNAI2 and CXCR4 were two putative miR-613 target genes upregulated in AF patients in whom miR-613 was downregulated, suggesting a potential regulatory relationship in the pathogenesis of AF. These novel findings indicate that miR-613 may serve as an important regulatory factor in AF progression and merit further investigation.

Nevertheless, some limitations must also be acknowledged. Due to the lack of detailed clinical metadata in publicly available datasets, adjustments for potential confounding factors such as age and clinical characteristics were not possible, which might affect the robustness of our results. Furthermore, the sample size, particularly for the mRNA dataset, was relatively small, which may limit the generalizability and stability of our conclusions. Although our qPCR validation experiments provided direct evidence supporting the bioinformatics predictions, additional in vivo experiments and validation in clinical tissue samples with larger cohorts are necessary to further elucidate the precise molecular roles of these miRNA-mRNA interactions in AF.

In conclusion, compared with patients of normal sinus rhythm, miRNA-613 was significantly down-regulated in patients suffering from atrial fibrillation by using bioinformatics analysis. Our qPCR validation further confirmed that SNAI2 and CXCR4 may be target genes of miRNA-613, and the expression level of these genes was significantly up-regulated in patients with atrial fibrillation, while miRNA-613 was significantly down-regulated in patients suffering from atrial fibrillation. Our findings may provide new ideas for clarifying the molecular mechanism of atrial fibrillation.

## Author contributions

**Conceptualization:** Zhenyu Zhai, Longlong Hu.

**Data curation:** Yiligong Qi.

**Validation:** Longlong Hu, Zumao Gan.

**Writing – original draft:** Yiligong Qi.

**Writing – review & editing:** Zhenyu Zhai, Longlong Hu, Zumao Gan.

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
