## [Decision Letter · Decision Letter 0]

PONE-D-24-48788Microarray-based analysis and characterization of microRNAs associated with atrial fibrillationPLOS ONE

Dear Dr. Zhai,

Thank you for submitting your manuscript to PLOS ONE. After careful consideration, we feel that it has merit but does not fully meet PLOS ONE’s publication criteria as it currently stands. Therefore, we invite you to submit a revised version of the manuscript that addresses the points raised during the review process.

**ACADEMIC EDITOR: ** methodological issues should be carefully addressed. Limitations should be discussed avoiding speculations and factual errors.

We look forward to receiving your revised manuscript.

Kind regards,

Vincenzo Lionetti, M.D., PhD

Academic Editor

PLOS ONE

If you are reporting a retrospective study of medical records or archived samples, please ensure that you have discussed whether all data were fully anonymized before you accessed them and/or whether the IRB or ethics committee waived the requirement for informed consent. If patients provided informed written consent to have data from their medical records used in research, please include this information.'

3. Please note that your Data Availability Statement is currently missing [the DOI/accession number of each dataset OR a direct link to access each database]. If your manuscript is accepted for publication, you will be asked to provide these details on a very short timeline. We therefore suggest that you provide this information now, though we will not hold up the peer review process if you are unable.

Additional Editor Comments (if provided):

Reviewers' comments:

Reviewer's Responses to Questions

**Comments to the Author**

1. Is the manuscript technically sound, and do the data support the conclusions?

Reviewer #1: Partly

Reviewer #2: Partly

2. Has the statistical analysis been performed appropriately and rigorously? 

Reviewer #1: No

Reviewer #2: Yes

3. Have the authors made all data underlying the findings in their manuscript fully available?

Reviewer #1: No

Reviewer #2: Yes

4. Is the manuscript presented in an intelligible fashion and written in standard English?

Reviewer #1: Yes

Reviewer #2: Yes

5. Review Comments to the Author

Reviewer #1: The study explores key miRNA-mRNA interactions in atrial fibrillation using a bioinformatic approach. While potentially interesting, there are significant shortcomings that affect the interpretability of the results. The methodology was inadequately explained, and there was a lack of reproducible analysis provided. For instance, it is unclear what methods were employed to correct the background and normalize the data. It is also uncertain whether any filters were applied to exclude probes with no expression or if quality control measures such as principal component analysis were conducted. Moreover, confounding corrections do not appear to have been addressed, and it seems that multiple testing corrections (e.g., false discovery rate) were not utilized. In the context of transcriptomics research, this oversight is unacceptable. Furthermore, the lack of independent validation and the small sample sizes of the investigated datasets undermine the validity of their findings.

Reviewer #2: This study, while simple in its approach, leads to important and meaningful conclusions.

I would like to clarify why I believe the manuscript is partially technically valid. To strengthen the conclusions, it would be important to validate the findings observed through bioinformatics analysis with additional techniques, such as IHC, among others. Nevertheless, as preliminary data for further studies, the current study is technically valid.

Additionally, the statistical analysis is somewhat rigorous, but I would recommend using adjusted p-values (p-value adj) < 0.05 (or even 0.10) when selecting differentially expressed miRNAs and mRNAs, rather than relying solely on p-values. A p-value of < 0.05 indicates that the adjusted p-value is likely below 0.05, ensuring that the statistical analysis is both rigorous and appropriate. It would be preferable to consistently use either FC (fold-change) or logFC (log fold-change) throughout the manuscript. In Table 1, logFC is provided, while in the text, FC is used with a threshold of >2 (FC>2). To avoid confusion for readers, I suggest using both terms, perhaps in the table, or consistently using only one of the two throughout the manuscript.

I also suggest expanding the discussion to provide more detail, particularly in explaining the significance of the statistically significant Gene Terms (for BP, CC, MF) in relationship of AF, and what they may represent (Figure 2).

Overall, the results are promising, and I believe it is crucial to validate these findings with additional techniques. Lastly, I recommend reviewing the use of abbreviations throughout the manuscript, ensuring that they are defined in brackets the first time they are mentioned, and thereafter avoiding repetition of the full term.I suggest checking English too, but it seems to be quite fluent. I recommend broadening the discussion on your results by comparing them with what has already been observed in AF or similar conditions. If you don't find any comparison articles (I don't think so) explain in great detail your hypothesis also at the level of molecular mechanisms. This would be very useful for those who will do new in vivo studies and using other techniques to validate this study. So it would give an important strength to this study.

6. PLOS authors have the option to publish the peer review history of their article (what does this mean? ). If published, this will include your full peer review and any attached files.

**Do you want your identity to be public for this peer review?** For information about this choice, including consent withdrawal, please see our Privacy Policy .

Reviewer #1: No

Reviewer #2: No

---

## [Author Response · Author response to Decision Letter 1]

28 Mar 2025

Point-to-point response

We thank all three reviewers for their insights and suggestions, which have helped us greatly improve the manuscript. Below is a point-to-point response embedded in the reviewer’s comments (in blue).

Response to Reviewers’ comments:

Reviewer(s)' Comments to Author:

Reviewer #1: The study explores key miRNA-mRNA interactions in atrial fibrillation using a bioinformatic approach. While potentially interesting, there are significant shortcomings that affect the interpretability of the results. The methodology was inadequately explained, and there was a lack of reproducible analysis provided. For instance, it is unclear what methods were employed to correct the background and normalize the data. It is also uncertain whether any filters were applied to exclude probes with no expression or if quality control measures such as principal component analysis were conducted. Moreover, confounding corrections do not appear to have been addressed, and it seems that multiple testing corrections (e.g., false discovery rate) were not utilized. In the context of transcriptomics research, this oversight is unacceptable. Furthermore, the lack of independent validation and the small sample sizes of the investigated datasets undermine the validity of their findings.

We sincerely appreciate your thorough review and valuable comments, which have significantly contributed to improving the quality of our manuscript.

We fully acknowledge that our initial submission lacked sufficient detail in describing the data preprocessing, normalization methods, and quality control measures, which could affect the interpretability and reproducibility of our findings. In response to your constructive feedback, we have carefully and thoroughly addressed these issues in the revised manuscript, particularly the methodological concerns you raised.

1. Methodological details and quality control measures:

In the Methods section, we have now provided a detailed description of the preprocessing procedures applied to the publicly available miRNA expression dataset (GSE68475) and mRNA expression dataset (GSE31821). Briefly, For the miRNA expression dataset (GSE68475) downloaded from the public GEO database, raw miRNA microarray data were initially processed using GeneSpring GX software. All raw intensity values lower than 1.0 were first adjusted to 1.0 to minimize interference from extremely low-expressed miRNAs. Subsequently, data normalization was performed by referencing the median expression levels of each miRNA across the 11 normal sinus rhythm (NSR) samples, thereby reducing systematic technical biases between arrays. Additionally, miRNAs with maximum raw signal intensities lower than 20 across all samples were filtered out, ensuring the reliability of the subsequent analyses. Furthermore, we performed principal component analysis (PCA) as an additional quality control measure, and the PCA results demonstrated clear separation between AF and NSR samples in two-dimensional principal component space, indicating good data quality and suitability for subsequent differential expression analyses (Figure1 to reviewer1. A). For another mRNA expression dataset (GSE31821) utilized in this study, the raw microarray data (CEL files) were imported into R software and processed using the Robust Multi-array Average (RMA) method, including background correction, quantile normalization, and probe summarization to generate gene-level expression values. PCA was also performed as a quality control measure, and the results clearly showed distinct clustering of AF and control samples, further confirming data quality and supporting the validity of subsequent analyses (Figure1 to reviewer1. B).

The specific revised Materials and Methods content added is as follows:

2.1 AF-related miRNA expression profiling

An AF-related miRNA microarray dataset (GSE68475) was downloaded from the NCBI GEO database (NCBI, http://www.ncbi.nlm.nih.gov/geo/). This dataset included miRNA expression levels in right atrial appendage samples collected from individuals with AF or normal sinus rhythm (NSR) undergoing open-heart surgery at Oita University Hospital. In total, this dataset included 10 AF patient samples and 11 NSR patient samples. Raw microarray data were preprocessed and analyzed using GeneSpring GX software. Briefly, raw data values below 1.0 were set to 1.0 to minimize potential interference from very low-expression miRNAs. Next, the data for each miRNA were normalized to the median expression level of that miRNA across the 11 NSR samples, eliminating inter-array systematic errors and technical bias. Following normalization, we applied additional filtering to retain only miRNAs with raw signal intensities of ≥20 in at least one sample, ensuring the reliability of subsequent analyses. Principal component analysis (PCA) was performed as a quality control measure, and results demonstrated clear separation between AF and NSR groups, supporting the suitability of the dataset for differential expression analysis.

2.2 Differentially expressed miRNA identification

Differentially expressed miRNAs between AF and NSR patient samples were identified using the R limma package. Briefly, moderated t-tests were performed to compare miRNA expression between the two groups, and P values obtained from these tests were subsequently adjusted for multiple comparisons using the Benjamini-Hochberg False Discovery Rate (FDR) method. MiRNAs with adjusted P-value < 0.05 were considered statistically significant and selected for further analyses.

2.4 Identification of overlapping differentially expressed miRNAs and mRNAs associated with AF

The GSE31821 dataset was downloaded from the GEO database and included auricular tissue biopsy samples from 2 control individuals and 4 AF patients from which RNA were extracted for Affymetrix microarray analysis. Raw microarray data were imported into R and preprocessed using the Robust Multi-array Average (RMA) method, including background correction to reduce background noise, quantile normalization to eliminate technical variability among arrays, and probe summarization to obtain gene-level expression values. Principal component analysis was performed as a quality control measure, revealing clear separation between AF and control samples, supporting data reliability for subsequent analyses.

2. Multiple testing correction:

Concerning multiple testing correction, as recommended, we have now implemented the Benjamini-Hochberg False Discovery Rate (FDR) method in our revised analysis. Differentially expressed miRNAs and mRNAs were re-selected based on the adjusted P-value threshold (p-value adj < 0.05), and these updated analytical results have been clearly included in the revised manuscript. Following this adjustment, we identified a slightly reduced number of differentially expressed miRNAs (33 significantly upregulated and 34 downregulated) (Figure2 to reviewer1. A). Importantly, our key miRNAs of interest (including miR-613 and miR-339-5p) remained significantly differentially expressed, demonstrating that our core conclusions are robust to this statistical refinement. Additionally, the set of differentially expressed mRNAs remained unchanged after adjusting the P-values, possibly due to the relatively small sample size limiting the impact of this correction (Figure2 to reviewer1. B).

3. Adjustment for potential confounding factors and independent experimental validation:

Regarding your concern about potential confounding factors, we did not perform adjustments for possible confounders such as age and other clinical characteristics, mainly due to the lack of sufficiently detailed metadata available from the public datasets used in this study. We fully understand and sincerely accept your criticism, acknowledging this limitation as an inherent constraint of our study design. We have explicitly clarified this limitation in the revised manuscript and committed to addressing this issue through more rigorous and comprehensive experimental designs in future studies, utilizing datasets with more complete clinical information to minimize potential confounding effects and enhance the robustness of our conclusions.

We fully recognize your concerns regarding the absence of independent validation experiments in our initial submission. To address this important limitation, we have now conducted qPCR validation experiments to independently verify the miRNA-mRNA regulatory relationships predicted by our bioinformatic analysis. Specifically, we performed qPCR experiments to validate the regulatory relationships between miR-613 and its predicted targets SNAI2 and CXCR4, as well as between miR-339-5p and its predicted target FHL1.

Our qPCR results demonstrated clear evidence supporting these interactions: inhibition of miR-339-5p significantly increased FHL1 expression (P<0.0001), whereas the miR-339-5p mimic significantly decreased its expression (P<0.01); inhibition of miR-613 significantly elevated the expression levels of SNAI2 (P<0.0001) and CXCR4 (P<0.01), while the miR-613 mimic significantly reduced their expression (SNAI2, P<0.01; CXCR4, P<0.001) (Figure3 to reviewer1. A-C).

These experimental data strongly validate our computational predictions and substantially enhance the robustness and interpretability of our findings. We have clearly presented these new experimental results in the revised manuscript (Figure 8 in the revised manuscript), and thoroughly discussed the implications in the revised Discussion section.

The newly added experimental validation section (Results 3.4) in the revised manuscript is:

3.4 Experimental validation of miRNA-mRNA interactions via quantitative PCR

To experimentally validate the predicted miRNA-mRNA regulatory relationships derived from bioinformatics analyses, quantitative polymerase chain reaction (qPCR) assays were performed. Specifically, we investigated the direct effects of miR-339-5p and miR-613 on the expression levels of their predicted target genes (FHL1, SNAI2, and CXCR4) in vitro (Figure 8).

The qPCR results demonstrated that the expression of FHL1 significantly increased when cells were treated with the hsa-miR-339-5p inhibitor compared to the inhibitor negative control (Inhibitor-NC) group (P < 0.0001), indicating a strong negative regulatory relationship. Conversely, treatment with the miR-339-5p mimic markedly decreased the FHL1 expression levels compared with the mimic negative control (Mimic-NC) group (P < 0.01). These findings clearly support the prediction that miR-339-5p negatively regulates FHL1, reinforcing the complexity of miRNA-mRNA interactions in atrial fibrillation.

Similarly, our analysis confirmed that miR-613 significantly modulated the expression of its putative target genes, SNAI2 and CXCR4. Inhibition of miR-613 led to a substantial upregulation of SNAI2 (P < 0.0001) and CXCR4 (P < 0.01), compared with the Inhibitor-NC group. In contrast, overexpression of miR-613 using a miR-613 mimic significantly decreased the expression of SNAI2 (P < 0.01) and CXCR4 (P < 0.001) compared with the Mimic-NC group. These experimental validations confirm the bioinformatics predictions and substantiate our hypothesis that the downregulation of miR-613 observed in atrial fibrillation patients could contribute to disease progression by relieving repression of SNAI2 and CXCR4 expression.

Collectively, these qPCR validation results provide robust experimental evidence confirming the predicted regulatory interactions and strengthen the mechanistic insights into miRNA involvement in atrial fibrillation pathogenesis.

The updated limitation section in the Discussion now states:

Nevertheless, some limitations must also be acknowledged. Due to the lack of detailed clinical metadata in publicly available datasets, adjustments for potential confounding factors such as age and clinical characteristics were not possible, which might affect the robustness of our results. Furthermore, the sample size, particularly for the mRNA dataset, was relatively small, which may limit the generalizability and stability of our conclusions. Although our qPCR validation experiments provided direct evidence supporting the bioinformatics predictions, additional in vivo experiments and validation in clinical tissue samples with larger cohorts are necessary to further elucidate the precise molecular roles of these miRNA-mRNA interactions in AF.

Again, we deeply appreciate your thorough review and insightful suggestions, and we have earnestly and diligently revised the manuscript. We hope our detailed revisions adequately address your concerns and resolve your doubts about the rigor of our data analysis.

Figure1 to reviewer1. Principal component analysis (PCA) of microarray datasets used in this study.

(A) PCA plot of the miRNA expression profiles from dataset GSE68475, showing clear separation between normal sinus rhythm (NSR, blue dots) and atrial fibrillation (AF, red dots) samples. Ellipses represent 90% confidence intervals for each group.

(B) PCA plot of the mRNA expression profiles from dataset GSE31821, demonstrating distinct grouping of AF patient samples (red dots) and control samples (blue dots), confirming suitable data quality for further analyses.

Figure2 to reviewer1. Volcano plots illustrating differential expression analyses for miRNAs and mRNAs.

(A) Volcano plot displaying the differentially expressed miRNAs identified between AF and NSR samples. Red dots indicate significantly upregulated miRNAs, green dots represent significantly downregulated miRNAs, and black dots represent miRNAs without significant changes.

(B) Volcano plot illustrating the differential expression of mRNAs between AF and control samples. Similarly, red and green dots indicate significantly upregulated and downregulated mRNAs, respectively.

Figure3 to reviewer1. Validation of miRNA-mRNA regulatory relationships by qPCR.

qPCR validation of miRNA-mRNA regulatory relationships in atrial fibrillation. Expression level of FHL1 (A), SNAI2 (B) and CXCR4 (C) following transfection with hsa-miR-613 inhibitor or mimic compared to negative controls. Data represent mean ± SD of three independent experiments. **P < 0.01; ***P < 0.001; ****P < 0.0001.

Reviewer #2: This study, while simple in its approach, leads to important and meaningful conclusions.

I would like to clarify why I believe the manuscript is partially technically valid. To strengthen the conclusions, it would be important to validate the findings observed through bioinformatics analysis with additional techniques, such as IHC, among others. Nevertheless, as preliminary data for further studies, the current study is technically valid.

Additionally, the statistical analysis is somewhat rigorous, but I would recommend using adjusted p-values (p-value adj) < 0.05 (or even 0.10) when selecting differentially expressed miRNAs and mRNAs, rather than relying solely on p-values. A p-value of < 0.05 indicates that the adjusted p-value is likely below 0.05, ensuring that the statistical analysis is both rigorous and appropriate. It would be preferable to consistently use either FC (fold-change) or logFC (log fold-change) throughout the manuscript. In Table 1, logFC is provided, while in the text, FC is used with a threshold of >2 (FC>2). To avoid confusion for readers, I suggest using both terms, perhaps in the table, or consistently using only one of the two throughout the manuscript.

I also suggest expanding the discussion to provide more detail, particularly in explaining the significance of the statistically significant Gene Terms (for BP, CC, MF) in relationship of AF, and what they may represent (Figure 2).

Overall, the results are promising, and I believe it is crucial to validate these findings with additional techniques. Lastly, I recommend reviewing the use of abbreviations throughout the manuscript, ensuring that they are defined in brackets the first time they are mentioned, and thereafter avoiding repetition of the full term.I suggest checking English too, but it seems to be quite fluent. I recommend broadening the discussion on your results by comparing them with what has al

---

## [Decision Letter · Decision Letter 1]

PONE-D-24-48788R1Microarray-based analysis and characterization of microRNAs associated with atrial fibrillationPLOS ONE

Dear Dr. Zhai,

Thank you for submitting your manuscript to PLOS ONE. After careful consideration, we feel that it has merit but does not fully meet PLOS ONE’s publication criteria as it currently stands. Therefore, we invite you to submit a revised version of the manuscript that addresses the points raised during the review process.

**ACADEMIC EDITOR: ** The revised form of the present manuscript suggest potential involvement of miRNA-613/ SNAI2/CXCR4 axis in particular myocardial phenotype. Additional Editor suggestions should be considered. 

We look forward to receiving your revised manuscript.

Kind regards,

Vincenzo Lionetti, M.D., PhD

Academic Editor

PLOS ONE

Journal Requirements:

Additional Editor Comments:

1) The revised manuscript highlights a new potential adaptive mechanism involving genes regulated by miRNA613 that deserve attention by the authors. Myocardial hibernation refers to a state of reduced contractile function and metabolism in viable human myocardium due to chronic hypoperfusion or dyssynchrony even if the heart is failing (please see J Cell Mol Med. 2014 Mar;18(3):396-414.;J Card Fail. 2009 Dec;15(10):920-8 ). In the setting of AF, particularly when persistent or accompanied by HF, ventricular myocardial hibernation may occur as an adaptive response to chronic hemodynamic stress and altered perfusion. In patients with AF, the downregulation of miRNA-613 leads to upregulation of SNAI2 and CXCR4, which in turn may contribute to fibrotic and structural remodeling (via SNAI2),enhanced cellular survival signaling and adaptation to chronic stress (via CXCR4), possible dedifferentiation of cardiomyocytes, reducing their metabolic activity and contractile function. These changes are consistent with a myocardial hibernation phenotype—where cells downregulate activity to preserve viability under chronic stress. Thus, the miRNA-613/ SNAI2/CXCR4 axis may act as a molecular switch or modulator of hibernation-like states in ventricular myocardium secondary to chronic AF. The authors are invited to include this perspective in their discussion including abovementioned articles.

2) Include in the title the miRNA-613/ SNAI2/CXCR4 axis.

Reviewers' comments:

Reviewer's Responses to Questions

**Comments to the Author**

1. If the authors have adequately addressed your comments raised in a previous round of review and you feel that this manuscript is now acceptable for publication, you may indicate that here to bypass the “Comments to the Author” section, enter your conflict of interest statement in the “Confidential to Editor” section, and submit your "Accept" recommendation.

Reviewer #2: (No Response)

2. Is the manuscript technically sound, and do the data support the conclusions?

Reviewer #2: Yes

3. Has the statistical analysis been performed appropriately and rigorously? 

Reviewer #2: Yes

4. Have the authors made all data underlying the findings in their manuscript fully available?

Reviewer #2: Yes

5. Is the manuscript presented in an intelligible fashion and written in standard English?

Reviewer #2: Yes

6. Review Comments to the Author

Reviewer #2: I believe that, with the methodological adjustments and revisions made, this study can now be considered valid and suitable for recommendation. The authors have clearly taken my comments into account and have implemented the requested changes appropriately.

Firstly, I appreciated their use of the adjusted p-value, even though some data were modified compared to the previous version. Nevertheless, the key significant results originally highlighted remain evident, thereby further supporting their hypothesis. This consistency demonstrates the robustness of their main conclusions in light of the refined statistical approach.

Moreover, I commend the authors for incorporating qPCR validation, especially as I had suggested using additional techniques to support the bioinformatics findings. This experimental validation has confirmed the observations derived from the bioinformatic analysis, adding substantial value to the study. These results meaningfully reinforce their computational predictions and significantly enhance the credibility and reliability of their conclusions.

Additionally, the revised discussion is more detailed and effectively underscores the authors’ central hypothesis. While in vivo validation would certainly be of interest for future studies, I believe that the analyses presented in this manuscript are sufficient for the current scope of the work.

7. PLOS authors have the option to publish the peer review history of their article (what does this mean? ). If published, this will include your full peer review and any attached files.

**Do you want your identity to be public for this peer review?** For information about this choice, including consent withdrawal, please see our Privacy Policy .

Reviewer #2: No

---

## [Author Response · Author response to Decision Letter 2]

17 Apr 2025

Point-to-point response

We thank all editors and reviewers for their insights and suggestions, which have helped us greatly improve the manuscript. Below is a point-to-point response embedded in the reviewer’s comments (in blue).

Response to Editors’ comments:

Additional Editor Comments:

1) The revised manuscript highlights a new potential adaptive mechanism involving genes regulated by miRNA613 that deserve attention by the authors. Myocardial hibernation refers to a state of reduced contractile function and metabolism in viable human myocardium due to chronic hypoperfusion or dyssynchrony even if the heart is failing (please see J Cell Mol Med. 2014 Mar;18(3):396-414.;J Card Fail. 2009 Dec;15(10):920-8 ). In the setting of AF, particularly when persistent or accompanied by HF, ventricular myocardial hibernation may occur as an adaptive response to chronic hemodynamic stress and altered perfusion. In patients with AF, the downregulation of miRNA-613 leads to upregulation of SNAI2 and CXCR4, which in turn may contribute to fibrotic and structural remodeling (via SNAI2),enhanced cellular survival signaling and adaptation to chronic stress (via CXCR4), possible dedifferentiation of cardiomyocytes, reducing their metabolic activity and contractile function. These changes are consistent with a myocardial hibernation phenotype—where cells downregulate activity to preserve viability under chronic stress. Thus, the miRNA-613/ SNAI2/CXCR4 axis may act as a molecular switch or modulator of hibernation-like states in ventricular myocardium secondary to chronic AF. The authors are invited to include this perspective in their discussion including abovementioned articles.

We sincerely appreciate your thoughtful comments, which have greatly improved the rigor and depth of our manuscript. According to your suggestion, we have expanded the discussion section to include the potential adaptive mechanism involving myocardial hibernation mediated by the miRNA-613/SNAI2/CXCR4 axis, referencing the recommended literature. Specifically, the newly added section is as follows:

Moreover, the downregulation of miRNA-613 and subsequent upregulation of SNAI2 and CXCR4 suggest that ventricular myocardium might enter a state analogous to myocardial hibernation. Myocardial hibernation refers to a protective remodeling process in which cardiac myocytes downregulate their contractile function and metabolic demand to maintain viability under conditions of chronic ischemia or hemodynamic stress. Typical characteristics include reduced myocardial contractility despite preserved cell viability, glycogen accumulation, decreased capillary density, downregulated metabolic enzyme activity, and activation of ischemia-related signaling pathways [62]. It has been reported that even in the absence of overt coronary ischemia, persistent rapid pacing and ventricular dyssynchrony can induce myocardial changes mimicking hibernation, such as increased glycogen storage and reduced expression of glycolytic enzymes, representing an adaptive response to sustained hemodynamic stress [63]. The transcriptional repressor SNAI2 promotes cellular survival under stress conditions, while elevated CXCR4 expression is considered a hallmark of protective myocardial responses under ischemic conditions. Therefore, the observed miRNA-613/SNAI2/CXCR4 axis might function as a molecular modulator facilitating a hibernation-like phenotype in ventricular myocardium under chronic AF-induced hemodynamic stress, ultimately protecting myocardial tissue from energetic depletion and stress-induced damage.

2) Include in the title the miRNA-613/ SNAI2/CXCR4 axis.

We sincerely appreciate for your valuable suggestion. We have revised the title of our manuscript to clearly highlight the miRNA-613/SNAI2/CXCR4 axis, as recommended. The revised title is as follows:

Microarray-based analysis reveals a novel role of the miRNA-613/SNAI2/CXCR4 axis in atrial fibrillation

Response to Reivewers’ comments:

Reviewer #2: I believe that, with the methodological adjustments and revisions made, this study can now be considered valid and suitable for recommendation. The authors have clearly taken my comments into account and have implemented the requested changes appropriately.

Firstly, I appreciated their use of the adjusted p-value, even though some data were modified compared to the previous version. Nevertheless, the key significant results originally highlighted remain evident, thereby further supporting their hypothesis. This consistency demonstrates the robustness of their main conclusions in light of the refined statistical approach.

Moreover, I commend the authors for incorporating qPCR validation, especially as I had suggested using additional techniques to support the bioinformatics findings. This experimental validation has confirmed the observations derived from the bioinformatic analysis, adding substantial value to the study. These results meaningfully reinforce their computational predictions and significantly enhance the credibility and reliability of their conclusions.

Additionally, the revised discussion is more detailed and effectively underscores the authors’ central hypothesis. While in vivo validation would certainly be of interest for future studies, I believe that the analyses presented in this manuscript are sufficient for the current scope of the work.

Thank you very much for your insightful suggestions and positive evaluation of our manuscript. Your detailed and constructive feedback has significantly enhanced the scientific rigor and clarity of our work. In future studies, we plan to further validate these findings through comprehensive in vivo experiments to deepen our understanding of the underlying molecular mechanisms of atrial fibrillation.

---

## [Editor Report · Decision Letter 2]

PONE-D-24-48788R2Microarray-based analysis reveals a novel role of the miRNA-613/SNAI2/CXCR4 axis in atrial fibrillationPLOS ONE

Dear Dr. Zhai,

Thank you for submitting your manuscript to PLOS ONE. After careful consideration, we feel that it has merit but does not fully meet PLOS ONE’s publication criteria as it currently stands. Therefore, we invite you to submit a revised version of the manuscript that addresses the points raised during the review process.

**ACADEMIC EDITOR: ** Whilst the authors have included data from qPCR experiments (Figure 8) in the most recent revision, there is no reporting of the qPCR methodology, including the samples used, their origin and the in vitro culture methods to generate them, all of which limit the reproducibility of this study should a reader wish to replicate the findings. As such, the manuscript doesn't currently meet PLOS publication criteria which requires that "Experiments, statistics, and other analyses are performed to a high technical standard and are described in sufficient detail."

The authors are strongly invited to update the reporting of their methodology. 

We look forward to receiving your revised manuscript.

Kind regards,

Vincenzo Lionetti, M.D., PhD

Academic Editor

PLOS ONE

---

## [Author Response · Author response to Decision Letter 3]

14 May 2025

Point-to-point response

We thank academic editor for insights and suggestions, which have helped us greatly improve the manuscript. Below is a point-to-point response embedded in the editor’s comments (in blue).

Response to Editor’s comments:

Editor(s)' Comments to Author:

ACADEMIC EDITOR: Whilst the authors have included data from qPCR experiments (Figure 8) in the most recent revision, there is no reporting of the qPCR methodology, including the samples used, their origin and the in vitro culture methods to generate them, all of which limit the reproducibility of this study should a reader wish to replicate the findings. As such, the manuscript doesn't currently meet PLOS publication criteria which requires that "Experiments, statistics, and other analyses are performed to a high technical standard and are described in sufficient detail."

The authors are strongly invited to update the reporting of their methodology.

Thank you for your constructive feedback. We apologize for the earlier omission of key methodological details. We have now added a detailed description of our qPCR procedures to the Materials and Methods section. Thank you again for your guidance in strengthening the scientific rigor and reproducibility of our manuscript.

The newly added content is as follows:

2.6 Experimental Model and Cell Culture

Human ventricular cardiomyocytes (AC16; Sigma-Aldrich, Cat# SCC-AC16) were cultured in DMEM/F-12 (Gibco, Cat# 11320-033) supplemented with 10 % fetal bovine serum (FBS; Gibco, Cat# 10099-141) and 1 % penicillin-streptomycin (Gibco, Cat# 15140-122) at 37 °C in a humidified atmosphere containing 5 % CO₂. Cells were maintained mycoplasma-free, authenticated by short-tandem-repeat profiling, and used between passages 4–10 to ensure phenotypic stability. For all experiments, cells were seeded at 2 × 10⁵ cells well⁻¹ in collagen-I-coated 6-well plates (Corning) and allowed to reach ~70 % confluence before transfection.

2.7 miRNA Mimic/Inhibitor Transfection

Gain- and loss-of-function studies were performed with synthetic hsa-miR-613 or hsa-miR-339-5p mimics, inhibitors, and matched negative controls (RiboBio, Guangzhou). Oligonucleotides were transfected at a final concentration of 50 nM using Lipofectamine 3000 (Thermo Fisher Scientific, Cat# L3000-015) according to the manufacturer’s instructions. Briefly, Lipofectamine and oligonucleotides were separately diluted in Opti-MEM (Gibco), combined (1:1 vol/vol), incubated for 15 min at room temperature, and added dropwise to the culture medium. Six hours later, the medium was replaced with fresh growth medium. Cells were harvested 48 h post-transfection for RNA extraction.

2.8 RNA Isolation and Quantitative Reverse Transcription PCR

Total RNA was extracted with TRIzol™ Reagent (Thermo Fisher Scientific, Cat# 15596-018) and quantified spectrophotometrically (NanoDrop 2000; Thermo Fisher). RNA integrity was verified by agarose-gel electrophoresis. For mRNA analysis, 500 ng RNA were reverse-transcribed with the PrimeScript™ RT Reagent Kit (Takara, Cat# RR037A) using random hexamers; for miRNA analysis, 200 ng RNA were reverse-transcribed with the Mir-X™ miRNA First Strand Synthesis Kit (Takara, Cat# 638313) employing stem-loop primers. Quantitative PCR was performed on a CFX96 Touch™ Real-Time PCR Detection System (Bio-Rad) with TB Green® Premix Ex Taq™ II (Takara, Cat# RR820A). The 20 µL reaction comprised 10 µL 2× TB Green mix, 0.4 µL of each 10 µM primer, 1 µL cDNA, and 8.2 µL nuclease-free water. Cycling parameters: 95 °C 30 s; 40 cycles of 95 °C 5 s and 60 °C 30 s; followed by melt-curve analysis. Primer sequences for GAPDH, FHL1, CXCR4, SNAI2, and U6 (miRNA reference) are listed in table shown below:

Primer sequences used for qPCR

Target Forward (5'→3') Reverse (5'→3')

GAPDH GGAAGCTTGTCATCAATGGAAATC TGATGACCCTTTTGGCTCCC

FHL1 TGCTGCCTGAAATGCTTTGAC GCCAGAAGCGGTTCTTATAGTG

CXCR4 GGGCAATGGATTGGTCATCCT TGCAGCCTGTACTTGTCCG

SNAI2 GAACTGGACACACATACAGTGATT GGCTGTATGCTCCTGAGCTG

2.9 Data Analysis

Ct values were averaged from three technical replicates per sample. Relative expression was calculated using the 2^-ΔΔCt method with GAPDH (mRNA) or U6 (miRNA) as internal controls and the corresponding negative-control group (Mimic-NC or Inhibitor-NC) as calibrator. Each experiment was repeated independently three times (n = 3). Data are presented as mean ± SD. Statistical significance between two groups was assessed by unpaired two-tailed Student’s t-test in GraphPad Prism 9; p < 0.05 was considered significant.

---

## [Editor Report · Decision Letter 3]

Microarray-based analysis reveals a novel role of the miRNA-613/SNAI2/CXCR4 axis in atrial fibrillation

PONE-D-24-48788R3

Dear Dr. Zhai,

We’re pleased to inform you that your manuscript has been judged scientifically suitable for publication and will be formally accepted for publication once it meets all outstanding technical requirements.

Kind regards,

Vincenzo Lionetti, M.D., PhD

Academic Editor

PLOS ONE
---

## [Editor Report · Acceptance letter]

PONE-D-24-48788R3

PLOS ONE

Dear Dr. Zhai,

I'm pleased to inform you that your manuscript has been deemed suitable for publication in PLOS ONE. Congratulations! Your manuscript is now being handed over to our production team.

Kind regards,

on behalf of

Prof. Vincenzo Lionetti

Academic Editor

PLOS ONE